# Enhanced Anti-Mold Property and Mechanism Description of Ag/TiO_2_ Wood-Based Nanocomposites Formation by Ultrasound- and Vacuum-Impregnation

**DOI:** 10.3390/nano10040682

**Published:** 2020-04-04

**Authors:** Lin Lin, Jiaming Cao, Jian Zhang, Qiliang Cui, Yi Liu

**Affiliations:** 1Key Laboratory of Wooden Materials Science and Engineering of Jilin Province, Beihua University, Jilin 132013, China; linlin_beihua@126.com (L.L.); 13381448456@163.com (J.C.); 2College of Science, Beihua University, Jilin 132013, China; 3State Key Laboratory of Superhard Materials, Jilin University, Changchun 130012, China; cql@jlu.edu.cn; 4College of Materials Science and Technology, Beijing Forestry University, Beijing 100083, China; liuyichina@bjfu.edu.cn

**Keywords:** Ag/TiO_2_, wood-based composites, ultrasound impregnation, vacuum impregnation

## Abstract

Ag/TiO_2_ wood-based nanocomposites were prepared by the methods of ultrasound impregnation and vacuum impregnation. The as-prepared samples were characterized by field emission scanning electron microscopy (FESEM), energy-dispersive spectroscopy (EDS), Fourier transform infrared spectroscopy (FTIR), mercury intrusion porosimetry (MIP), and water contact angles (WCAs). The anti-mold properties of the Ag/TiO_2_ wood-based nanocomposites were improved by 14 times compared to those of the original wood. The nano-Ag/TiO_2_, which was impregnated in the tracheid and attached to the cell walls, was able to form a two-stage rough structure and reduce the number of hydroxyl functional groups on the wood surfaces. The resulting decline of wood hydrophobic and equilibrium moisture content (EMC) destroyed the moisture environment necessary for mold survival. Ag/TiO_2_ was deposited in the wood pores, which reduced the number and volume of pores and blocked the path of mold infection. Thus, the anti-mold properties of the Ag/TiO_2_ wood-based nanocomposite were improved by cutting off the water source and blocking the mold infection path. This study reveals the anti-mold mechanism of Ag/TiO_2_ wood-based nanocomposites and provides a feasible pathway for wood-based nanocomposites with anti-mold functions.

## 1. Introduction

Wood is widely used in many applications due to its excellent material properties and renewable nature. However, wood is susceptible to mold infection in wet environments. Dense mold spots not only affect the aesthetic and decorative functions of wood but also reduce its mechanical properties [1,2]. Long-term contact with moldy wood and inhalation of its spores can cause allergies, respiratory inflammation, and mucosal disease [3,4,5]. In recent decades, the study of anti-mold treatments for wood has attracted much attention [6,7,8,9,10]. Compared with traditional organic anti-mold agents, inorganic agents have attracted more attention due to their unique features, such as earth-abundance, chemical-stability, and low-cost [11,12]. Nano-Ag/TiO_2_ is recognized as one of the best inorganic anti-mold agents [13,14]. Ag nanoparticles can significantly enhance the photocatalytic activity of TiO_2_ by restraining the recombination of electron-hole pairs, which can lead to the improvement of the anti-mold effect of TiO_2_ under natural light, weak light, and dark conditions [15,16]. Furthermore, nano-Ag has synergistic anti-mold effects with TiO_2_ due to its own anti-mold abilities [17]. However, after doping with Ag, the enhanced polarity of TiO_2_ makes it easier to agglomerate [18,19,20].

In this work, to obtain anti-mold wood-based composites, sodium hexametaphosphate (SHMP) and γ-(2,3-epoxypropoxy)propytrimethoxysilane (KH560) were used as composite modifiers to improve the dispersion of Ag/TiO_2_ nanoparticles, and the impregnation depth of Ag/TiO_2_ nanoparticles in wood was increased by means of ultrasonic-assisted impregnation and vacuum-assisted impregnation. Wood mold is a complicated fungal infection process closely related to the porosity and moisture of wood. Therefore, the anti-mold properties of Ag/TiO_2_ wood-based nanocomposites were tested and characterized by scanning electron microscope (SEM), energy-dispersive X-ray spectroscopy (EDS), Fourier transform infrared spectroscopy (FTIR), mercury intrusion porosimetry (MIP), and water contact angles (WCAs).

## 2. Materials and Methods

### 2.1. Materials

All chemical materials were supplied by Beijing Chemical Company Limited (Beijing, China). Wood specimens were obtained from the sapwood portions of mongolica wood (*Pinus sylvestris* var. *mongolica*), which is a fast-growing tree species in northeastern China. All specimens were free of mold, blue stains, insect boring marks, and knots. The wood blocks for the anti-mold performance test were processed into a shape that was 50 mm (longitudinal) × 20 mm (tangential) × 5 mm (radial) in size; the others were 20 mm × 20 mm × 20 mm.

### 2.2. Synthesis

The Ag/TiO_2_ wood-based nanocomposites were prepared via ultrasonic impregnation and vacuum impregnation.

#### 2.2.1. Surface Modification of the Ag/TiO_2_ Nanoparticles

We mixed 0.6 g of SHMP and 2 g of Ag/TiO_2_ nanoparticles (anatase-type, average size: 30 nm, mass fraction of silver: 1%) in 100 mL of deionized water under gentle stirring at room temperature. During the stirring process, an alcohol solution containing 20% KH560 was added dropwise into the mixed solution until the mass ratio of Ag/TiO_2_ and KH560 was 20:1.

#### 2.2.2. Ultrasound Impregnation

Wood specimens were immersed in the solution for 20 min under ultrasonic oscillation at a power of 150 W. Then, the specimens were dried in a vacuum oven at 40 °C. The wood sample impregnated with Ag/TiO_2_ nanoparticles via ultrasound is abbreviated as UW.

#### 2.2.3. Vacuum Impregnation

The wood specimens were immersed in the solution for 20 min under a −0.04 MPa vacuum. Then, the specimens were dried in a vacuum oven at 40 °C. The wood sample impregnated with Ag/TiO_2_ nanoparticles via vacuum assistance is abbreviated as VW.

### 2.3. Anti-Mold Test

Anti-mold properties were evaluated according to the Chinese Standard GB/T 18261-2000, the testing method for anti-mold chemicals in controlling mold and blue stain fungi on wood [21]. Thirty wood samples of each type were steeped in deionized water for 48 h to ensure that the samples contained enough water for mold growth. The samples were sealed in a glass container under natural light for 30 days; an environmental condition of 90% humidity (implemented by KNO_3_) and 20 °C was maintained. Mold control effectiveness (*M*) was calculated according to the equation as follows:$$M=\frac{N}{T}\times 100\%$$
where *N* is the number of samples with no obvious mold or blue-stain fungi on the surface (<5% of the total surface area) and *T* is the number of tested samples.

### 2.4. Characterization

The morphology of the Ag/TiO_2_ wood-based nanocomposites was observed by field emission scanning electron microscopy (FESEM, SU8010, Hitachi Co., Tokyo, Japan). The chemical compositions of the samples at different depths from the surface (at axial depths of 5 mm, 10 mm, 15 mm, and 20 mm) were determined by energy-dispersive spectroscopy (EDS, attached to the FESEM). The chemical changes of the samples were recorded by Fourier transform infrared spectroscopy (FTIR, Avatar 330, Nicolet Co., Madison, WI, USA). The pore-size distribution of the samples was identified by the mercury intrusion method (MI, Autopore IV 9510, Micromeritics Instrument Co., Norcross, GA, USA). The water contact angles (WCAs) at ambient temperature were measured on an contact angle system (WCA, OCA 20, Dataphysics Co., Filderstadt, Germany). An average of three measurements selected from the transverse section, radial section, and chord section of each sample was applied to calculate the final WCAs value every five seconds. The equilibrium moisture content (EMC) was measured in a temperature and humidity chamber (SNKC-150L, Snugen Co., Tokyo, Japan).

## 3. Results and Discussion

### 3.1. Anti-Mold Properties

The mold growth on the surface of the original wood, UW, and VW is shown in Figure 1. The surface of the original wood was completely covered by gray-green mycelia, some of which produced obvious plaque and had lost their natural color. In contrast, UW and VW retained their original color and few samples exhibited discolorations. The mold control effectiveness of UW and VW was 93% and 97%, which was improved by 13 times and 14 times, respectively, as compared with the original wood (Table 1).

### 3.2. Field Emission Scanning Electron Microscopy (FESEM) Analysis

The distribution and morphology of the original wood and treated wood are shown in Figure 2. As shown in the FESEM images of the transverse sections of UW and VW, abundant spherical Ag/TiO_2_ nanoparticles presented on the cell wall of the wood, which made the cell wall uneven [22,23].

### 3.3. Energy-Dispersive Spectroscopy (EDS) Analysis

The chemical elemental compositions of the original wood and the Ag/TiO_2_ wood-based nanocomposites were determined via EDS, and the results are shown in Figure 3. The element Ti was clearly detected, which confirmed that Ag/TiO_2_ was successfully deposited onto the surface of UW and VW. According to the Ti elemental mass percentage, the mass percentage of Ag/TiO_2_ on the UW surface was calculated to be 15.33%, while the mass percentage of Ag/TiO_2_ on the VW surface was 15.23%.

### 3.4. Distribution of Ag/TiO_2_

Anti-mold wood might be cut from the surface material during subsequent processing into wood products. Therefore, by exploring the impregnation depth and distribution of Ag/TiO_2_ nanoparticles in wood, we can determine the anti-mold properties of Ag/TiO_2_ wood-based nanocomposites in practical application. Figure 4 shows the mass fraction of Ag/TiO_2_ at the longitudinal depth of the wood. The longitudinal distribution of Ag/TiO_2_ in the different woods was quite different. A large number of nanoparticles were concentrated on the wood surfaces. With an increase in longitudinal depth, the mass fraction of Ag/TiO_2_ decreased. At 20 mm away from the surface, the mass fraction of Ag/TiO_2_ in UW was 1.05%, and that of VW was 0.97%. The minimum anti-mold concentration of Ag/TiO_2_ was 0.125%. At a depth of 20 mm, the concentration of Ag/TiO_2_ was eight times higher than that of the minimum anti-mold concentration and still had good anti-mold properties [24,25]. Therefore, processing had little impact on the mold-proof performance and anti-mold properties of the wood.

### 3.5. Mercury Intrusion Porosimetry (MIP) Analysis

The pore structure of wood is vital to prevent mold because mold attacks wood by entering the pores. As shown in Table 2, *Pinus sylvestris* was classified into four pore-size classes: two classes of macropores (pore radii 2–58 and 0.5–2 µm), mesopores (80–500 nm), and micropores (3.6–80 nm) [26]. 

The pore conditions of the original wood, UW, and VW are shown in Table 3. The porosities of UW and VW were 61.48% and 56.90% and were reduced by 13.24% and 19.70%, respectively, compared to the original wood. The decreased pore volume and increased average pore diameter are attributed to the reduced number of micropores and mesopores in UW and VW, indicating that the deposits of Ag/TiO_2_ nanoparticles were mainly supported on pores of this size.

The total volume of pores vs. the pore diameter of the original wood, UW, and VW is presented in Figure 5. The number of pores was equivalent when the pore diameters were greater than 36,642 nm. In a pore diameter range of 36,642–5019 nm, the increase in the total volume of pores of VW was similar to that of the original wood, while that of UW was significant. This pore diameter size corresponded to the size of the resin duct and pit chamber (Table 2). This may be due to the ultrasonic “acoustic cavitation” phenomenon, which can promote the loss of resin and result in an increase in the total volume of pores. In a pore diameter range of 5019–349 nm, the increase in the total volume of pores in UW and VW was greater than that of the original wood. This may be due to the vacuum treatment and ultrasonic treatment producing a pressure differential that damaged the pit, resulting in an increase in the total volume of pores. In a pore diameter range of 349–77 nm, the increase of the total volume of pores in VW and UW was significantly less than that of the original wood. This pore diameter size corresponded to the size of the cytoderm and the distribution range of the nano-Ag/TiO_2_ particle size. It is speculated that the volume change was caused by the attachment of Ag/TiO_2_ to the cytoderm [27].

Figure 6 shows the differential pore volume vs. the pore diameter of the original wood, UW, and VW. The differential pore volume can reflect the distribution of the pore size. There was not a large difference in the number of pores between UW and VW when the pore diameter was greater than 316 nm. When the pore diameter was less than 316 nm, the pores of the original wood were significantly larger than those of UW and VW, indicating that the modified treatment significantly reduced the number of micropores and had little effect on the mesopores and macropores. After modification of the wood by Ag/TiO_2_, the size and volume of the pore diameters was reduced, which effectively closed the channels that allowed mold and spores to enter the wood and prevented further infection by mold (Figure 7).

### 3.6. Water Contact Angle (WCA) Analysis

The wettability of the original wood, UW, and VW was characterized by exploring the contact angle through a sessile drop on the samples. In Figure 8, the original wood surfaces exhibited hydrophilicity with a water contact angle of 91°, which is attributed to the abundant hydroxyl groups in the basic wood material. However, after subsequent modification by Ag/TiO_2_, the water contact angles of UW and VW reached 126° and 122°, respectively, indicating the emergence of hydrophobicity on initially hydrophilic wood surfaces. The hydrophobicity of UW and VW was mainly due to the formation of a nano-Ag/TiO_2_ two-stage rough structure on the wood surfaces, resulting in a “snowball effect” that enhanced the hydrophobicity of the composite material [28].

Figure 9 shows the water contact angles of the original wood, UW, and VW varying over time. The water contact angle of the original wood decreased the fastest. At 4.5 s, the contact angle was 13° and then less than 10°, indicating that the water was gradually spread upon and absorbed by the wood. The contact angle of the nanocomposites decreased slowly. At 60 s, the contact angles of UW and VW were 71° and 81°, respectively, indicating that the nanocomposites exhibited better continuous waterproof performance.

Water is the main component of mycobacterium and the medium by which fungi invade wood. Most fungi are able to grow when the moisture content of wood is 35–60%, but when the moisture content is less than 20%, the growth of fungi is inhibited. The equilibrium moisture content (EMC) of the original wood, UW, and VW are shown in Table 4. The results show the EMC of UW and VW was lower than that of original wood under the same humidity conditions, indicating the hydrophobic performance of UW and VW was significantly improved and the low moisture content of the wood-based nanocomposites was able to improve the anti-mold effect. The Ag/TiO_2_ nano-hydrophilic hydroxyl on wood fibers was associated with hydrogen bonding; the decrease of hydroxyl functional groups promotes the increase of wood hydrophobicity. Meanwhile, as shown in Figure 10, KH560 in the compound modifier introduced hydrophobic base head alkanes to further prevent water from entering the wood [29,30].

### 3.7. Fourier Transform Infrared Spectroscopy (FTIR) Analysis

Figure 11 shows the FTIR spectra of the original wood, UW, and VW. In the UW and VW spectra, the characteristic bands of Ti–O were observed at 770–500 cm^−1^ and 565 cm^−1^. The band of cellulose hydroxyl (–OH) at 3396 cm^−1^ became weaker and broader and shifted to larger wavenumbers of 3415 cm^−1^ and 3421 cm^−1^, respectively, indicating the formation of hydrogen bonds between TiO_2_ and hydroxyl groups on the wood surface [31]. In addition, the bands at 1605 cm^−1^ (C=O), 1030 cm^−1^ (C=O), and 1233 cm^−1^ (phenolic hydroxyl) became weak due to the formation of coordination bonds between the carboxyl and phenolic hydroxyl groups in the wood and Ti^4+^ that formed on the surface of the Ag/TiO_2_. The characteristic bands of Si–O–Si in KH560 were observed at 1114 cm^−1^ and 1063 cm^-1^. Compared to UW, the band strength of VW at 770–500 cm^−1^ (Ti–O) was higher, indicating that vacuum impregnation was more favorable to support Ag/TiO_2_ nanoparticles on wood than ultrasound impregnation [32].

## 4. Conclusions

In summary, Ag/TiO_2_ wood-based nanocomposites with anti-mold functions were successfully prepared via ultrasound impregnation and vacuum impregnation. Nano-Ag/TiO_2_ can form a two-stage rough structure on wooden surfaces and introduce long-chain alkanes to make the wood hydrophobic, thus destroying the moist environment in wood that allows mold survival. At the same time, Ag/TiO_2_ was deposited in the wood pores, which reduced the number and overall volume of the pores and blocked the path of mold infection. This study revealed the anti-mold mechanism of Ag/TiO_2_ wood-based nanocomposites from the perspectives of water content and infection path and has potentially provided a feasible pathway for wood-based nanocomposites with anti-mold functions.

## Figures and Tables

**Figure 1 nanomaterials-10-00682-f001:**
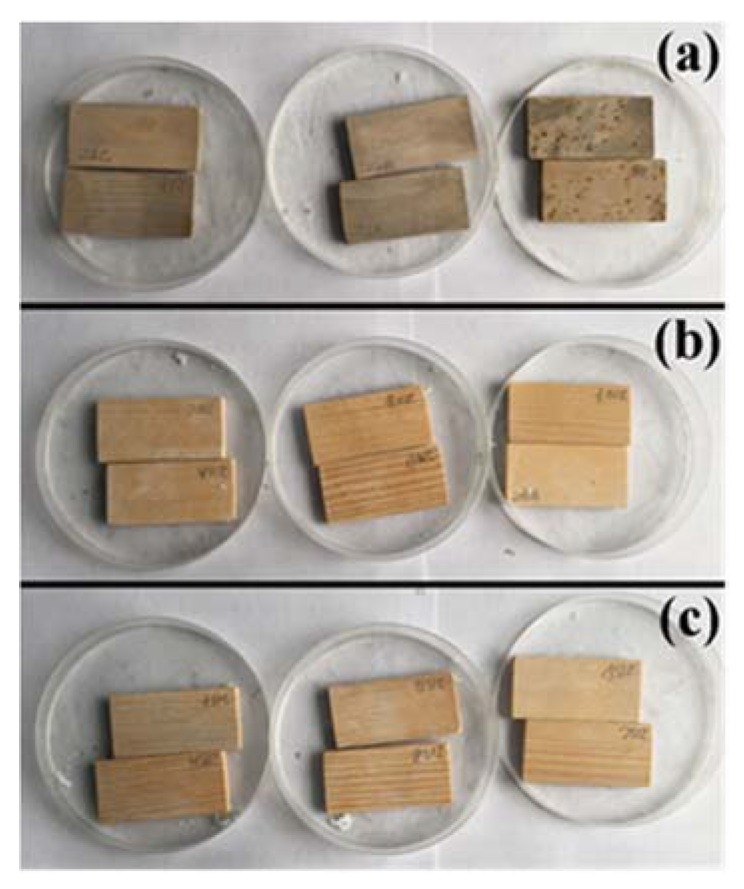
Mold infection of (**a**) original wood, (**b**) wood sample impregnated with Ag/TiO_2_ nanoparticles via ultrasound (UW), and (**c**) wood sample impregnated with Ag/TiO_2_ nanoparticles via vacuum assistance (VW).

**Figure 2 nanomaterials-10-00682-f002:**
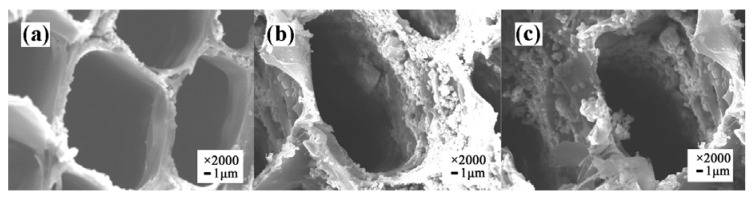
Field emission scanning electron microscopy (FESEM) of (**a**) original wood, (**b**) UW, and (**c**) VW.

**Figure 3 nanomaterials-10-00682-f003:**
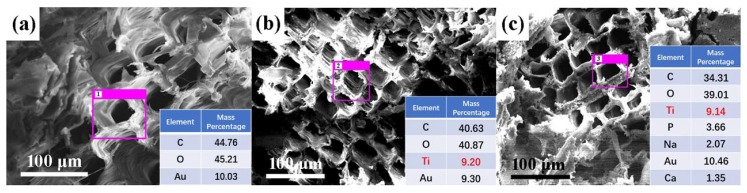
Energy-dispersive spectroscopy (EDS) of (**a**) original wood, (**b**) UW, and (**c**) VW. Data pertain to the selected area within the pink boxes.

**Figure 4 nanomaterials-10-00682-f004:**
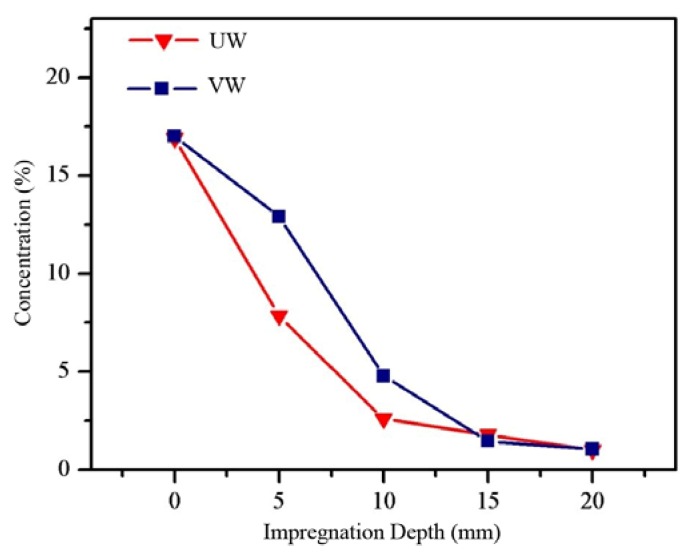
Longitudinal distribution of nano-Ag/TiO_2_ wood-based composite materials.

**Figure 5 nanomaterials-10-00682-f005:**
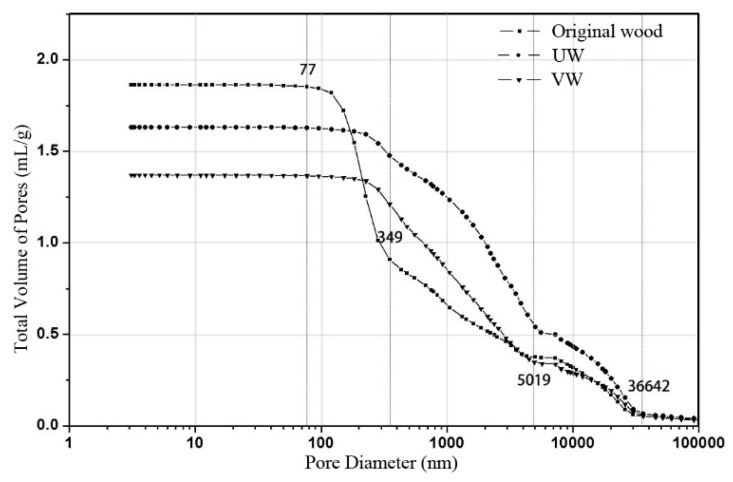
Total volume of pores vs. the pore diameter of original wood, UW, and VW.

**Figure 6 nanomaterials-10-00682-f006:**
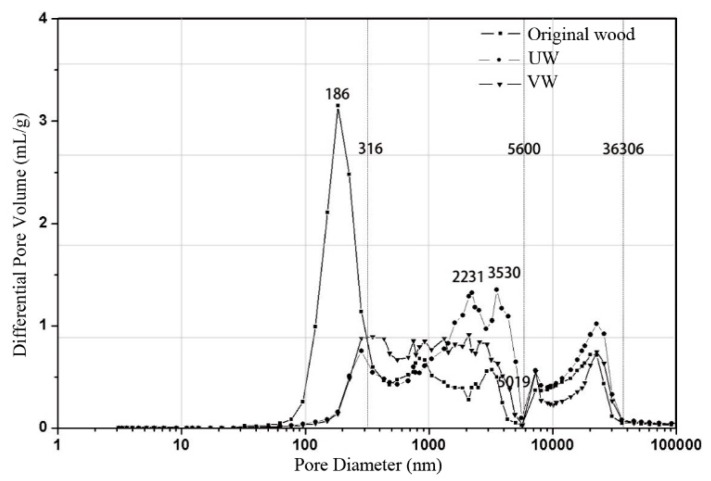
Differential pore volume vs. pore diameter of original wood, UW, and VW.

**Figure 7 nanomaterials-10-00682-f007:**
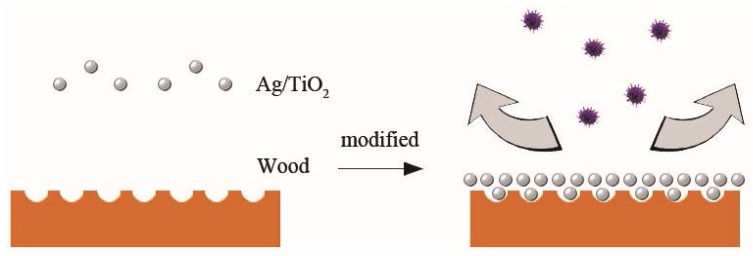
Nano-Ag/TiO_2_ prevented wood infection by mold.

**Figure 8 nanomaterials-10-00682-f008:**
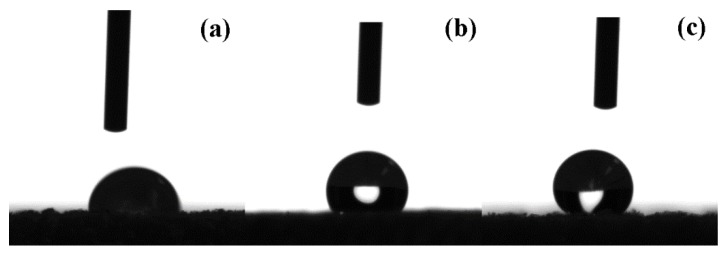
Water contact angles of (**a**) original wood, (**b**) UW, and (**c**) VW.

**Figure 9 nanomaterials-10-00682-f009:**
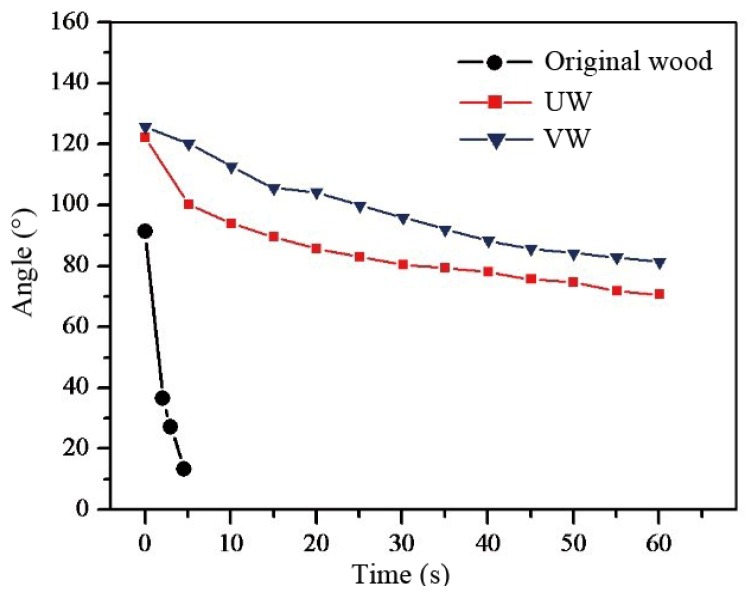
Dynamic water contact angle of specimens.

**Figure 10 nanomaterials-10-00682-f010:**
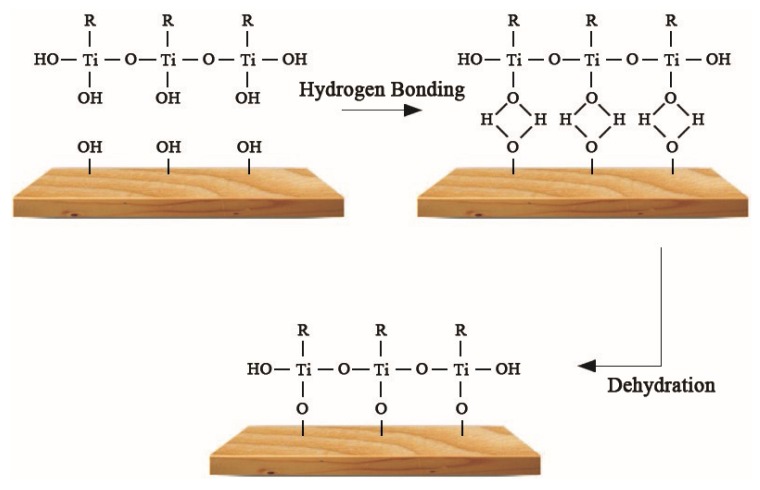
γ-(2,3-epoxypropoxy)propytrimethoxysilane (KH560) bonding the wood surfaces.

**Figure 11 nanomaterials-10-00682-f011:**
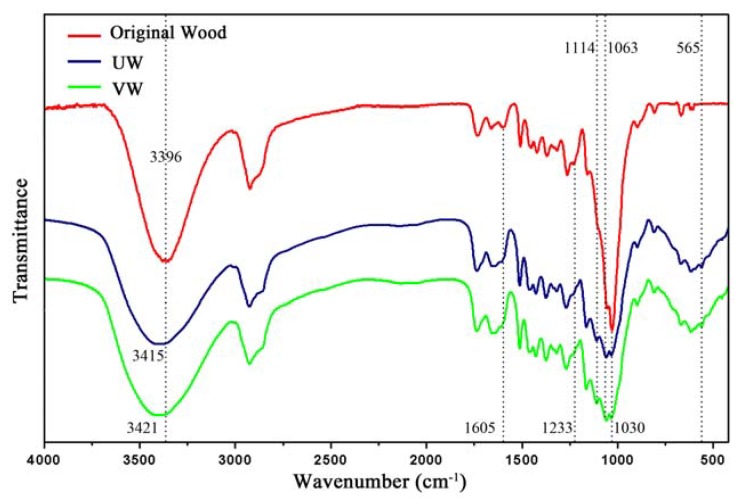
Fourier transform infrared spectroscopy (FTIR) of original wood, UW, and VW.

**Table 1 nanomaterials-10-00682-t001:** The effectiveness of mold control.

Sample	Original Wood	UW ^a^	VW ^b^
**Mold control effectiveness**	7%	93%	97%

^a^ Wood sample impregnated with Ag/TiO_2_ nanoparticles via ultrasound. ^b^ Wood sample impregnated with Ag/TiO_2_ nanoparticles via vacuum assistance.

**Table 2 nanomaterials-10-00682-t002:** Classification of pore structure in *Pinus sylvestris*.

Structure	Diameter Size	Pore Size Class
Tracheid	15–40 μm	Macropore
Resin duct	50–300 μm	Macropore
Pit chamber	4–30 μm	Macropore
Pit aperture	400 nm–6 μm	Macropore
Pit membrane	10 nm–8 μm	Micropore/Mesopore/Macropore
Cytoderm	2–100 nm	Micropore/Mesopore
Microfilament clearance	2–4.5 nm	Micropore

**Table 3 nanomaterials-10-00682-t003:** Pore conditions of original wood, UW, and VW.

Sample	Total Volume of Pore (mL/g)	Average Pore Diameter (nm)	Porosity (%)
Original wood	1.86	332.8	70.86
UW	1.62	1139.9	61.48
VW	1.37	708.6	56.90

**Table 4 nanomaterials-10-00682-t004:** Equilibrium moisture content of original wood, UW, and VW.

Relative Humidity	Sample	20%	30%	40%	50%	60%	70%	80%	90%
Equilibrium moisture content (%)	Original wood	5.4	5.6	7.1	8.9	10.7	12.5	15.6	19.8
	UW	5.0	5.2	5.3	5.3	6.1	7.6	10.1	12.1
	VW	4.8	4.9	5.0	5.2	5.8	7.1	9.7	11.6

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
