# Peer review of "Enhanced Anti-Mold Property and Mechanism Description of Ag/TiO2 Wood-Based Nanocomposites Formation by Ultrasound- and Vacuum-Impregnation"

_nanomaterials, 2020, doi:10.3390/nano10040682_

Round 1
Reviewer 1 Report
In my opinion this is a very interesting paper with a high degree of novelty. The two impregnation methods make the obtaining of the materials facile and reproductible and the characterization methods clearly sustain the successful accomplishment of the scope of the conducted research. I consider it worthy of publication in Nanomaterials, after the following Minor Revisions have been met:
1. Introduction needs improved. On Page 1, lines 36-37, authors state: "In recent decades, the study of anti-mildew treatments for wood has attracted much attention [6-8]", with only three references based strictly on metal oxide nanoparticles. There is a lot of recent research in the field of anti-mildew and wood antifungal wood protective coatings reported, not limited to only such type of coatings. I recommend that the authors cite the following recent works, as references [9, 10]:
L. Rosu, C.–D. Varganici, F. Mustata, T. Rusu, D. Rosu, I. Rosca. N. Tudorachi, C.–A. Teaca. Enhancing the Thermal and Fungal Resistance of Wood Treated with Natural and Synthetic Derived Epoxy Resins. ACS Sustainable Chem. Eng. 2018, 6(4), 5470–5478.
L. Rosu, F. Mustata, C.–D. Varganici, D. Rosu, T. Rusu, I. Rosca. Thermal behaviour and fungi resistance of composites based on wood and natural and synthetic epoxy resins cured with maleopimaric acid. Polym. Degrad. Stab. 2019, 160, 148–161.
2. Page 4, in Figure 2 it would be better if authors also included the ESEM micrograph of the initial non-coated wood in order to highlight the anti-mildew effect. The same in Figure 3: include the EDS for initial non treated wood.
3. Page 8, FTIR analysis: authors should mark in the figure all peak values discussed in the text, since it is hard for the reader to follow the discussion.
Author Response
Please see the attachment.
In response to the reviewer’s comments and suggestions, we have revised our manuscript carefully. We sincerely hope that this revised manuscript resolves all issues and can be considered for publication. Please find in attachment our detailed response to all issues point by point.
Thank you very much for your time and expertise in considering this work.

Reviewer 2 Report
Please see the attached file. The quality of English written language should be improved. There are in the paper some assumptions that are by my opinion not completely experimentally supported / confirmed.

Author Response

(The authors gave the same response as above.)

Round 2
Reviewer 2 Report
Dear Authors, thank you very much for your response to my comments, explanations and corrections. I do not have any additional requirements.